# Romanization-based Large-scale Adaptation of Multilingual Language Models

**Sukannya Purkayastha[1], Sebastian Ruder[2], Jonas Pfeiffer[2], Iryna Gurevych[1], Ivan Vulić[3]**

[1] Ubiquitous Knowledge Processing Lab,
Department of Computer Science and Hessian Center for AI (hessian.AI),
Technical University of Darmstadt
[2] Google DeepMind
[3] Language Technology Lab, University of Cambridge
www.ukp.tu-darmstadt.de

## Abstract

Large multilingual pretrained language models (mPLMs) have become the *de facto* state of the art for cross-lingual transfer in NLP. However, their large-scale deployment to many languages, besides pretraining data scarcity, is also hindered by the increase in vocabulary size and limitations in their parameter budget. In order to boost the capacity of mPLMs to deal with low-resource and unseen languages, we explore the potential of leveraging *transliteration* on a massive scale. In particular, we explore the UROMAN transliteration tool, which provides mappings from UTF-8 to Latin characters for all the writing systems, enabling inexpensive *romanization* for virtually any language. We first focus on establishing how UROMAN compares against other language-specific and manually curated transliterators for adapting multilingual PLMs. We then study and compare a plethora of data- and parameter-efficient strategies for adapting the mPLMs to romanized and non-romanized corpora of 14 diverse low-resource languages. Our results reveal that UROMAN-based transliteration can offer strong performance for many languages, with particular gains achieved in the most challenging setups: on languages with unseen scripts and with limited training data without any vocabulary augmentation. Further analyses reveal that an improved tokenizer based on romanized data can even outperform non-transliteration-based methods in the majority of languages.

## 1 Introduction

Massively multilingual language models (mPLMs) such as mBERT (Devlin et al., 2019) and XLM-R (Conneau et al., 2020) have become the driving force for a variety of applications in multilingual NLP (Ponti et al., 2020; Hu et al., 2020; Moghe et al., 2023). However, guaranteeing and maintaining strong performance for a wide spectrum of low-resource languages is difficult due to two crucial problems. The first issue is the *vocabulary*

| Language | Original Text | Romanized Text | Translation |
|---|---|---|---|
| Bhojpuri | जॉर्जियन भासा | jorjiyan bhaasaa | Georgian language |
| Sinhala | ග්‍රහලෝක | grahalooka | planets |
| Sindhi | ایران | ayran | Iran |
| Khmer | សេដ្ឋកិច្ច. | sedtthakicca | economy |

Figure 1: Romanization across different languages.

*size*, as the vocabulary is bound to increase with the number of languages added if per-language performance is to be maintained (Hu et al., 2020; Artetxe et al., 2020; Pfeiffer et al., 2022). Second, pretraining mPLMs with a *fixed model capacity* improves cross-lingual performance up to a point after which it starts to decrease; this is the phenomenon termed the *curse of multilinguality* (Conneau et al., 2020).

Transliteration refers to the process of converting language represented in one writing system to another (Wellisch et al., 1978). Latin script-centered transliteration or *romanization* is the most common form of transliteration (Lin et al., 2018; Amrhein and Sennrich, 2020; Demirsahin et al., 2022) as the Latin/Roman script is by far the most widely adopted writing script in the world (Daniels and Bright, 1996; van Esch et al., 2022).[1] Adapting mPLMs via transliteration can address the two aforementioned critical issues. **1)** Since the Latin script covers a dominant portion of the mPLM's vocabulary (e.g., 77% in case of mBERT, see Ács), 'romanizing' the remaining part of the vocabulary might mitigate the vocabulary size issue and boost vocabulary sharing. **2)** Since no new tokens are added during the romanization process, reusing pretrained embeddings from the mPLM's embedding matrix helps reuse the information already present within the mPLM, thereby allocating the model's parameter budget more efficiently.

However, the main drawback of transliteration seems to be the expensive process of creating effective language-specific transliterators, as they

---

[1] According to Encyclopedia Britannica, up to 70% of the world population is employing the Latin script.

typically require language expertise to curate dictionaries that map tokens from one language and script to another. Therefore, previous attempts at mPLM adaptation to unseen languages via transliteration (Muller et al., 2021; Chau and Smith, 2021; Dhamecha et al., 2021; Moosa et al., 2023) were constrained to a handful of languages due to the limited availability of language-specific transliterators, or were applied only to languages that have 'language siblings' with developed transliterators.

In this work, unlike previous work, we propose to use and then evaluate the usefulness of a universal romanization tool, UROMAN (Hermjakob et al., 2018), for quick, large-scale and effective adaptation of mPLMs to low-resource languages. The UROMAN tool disposes of language-specific curated dictionaries and maps any UTF-8 character to the Latin script, increasing the portability of romanization, with some examples in Figure 1.

We analyze language adaptation on a massive scale via UROMAN-based romanization on a set of 14 diverse low-resource languages. We conduct experiments within the standard parameter-efficient adapter-based cross-lingual transfer setup on two tasks: Named Entity Recognition (NER) on the WikiANN dataset (Pan et al., 2017; Rahimi et al., 2019), and Dependency Parsing (DP) with Universal Dependencies v2.7 (Nivre et al., 2020). Our key results suggest that UROMAN-based transliteration can offer strong performance on par or even outperforming adaptation with language-specific transliterators, setting up the basis for wider use of transliteration-based mPLM adaptation techniques in future work. The gains with romanization-based adaptation over standard adaptation baselines are particularly pronounced for languages with unseen scripts (~8-22 performance points) without any vocabulary augmentation.[2]

## 2 Background

**Why UROMAN-Based Romanization?** UROMAN-based romanization is not always fully reversible, and its usage for transliteration has thus been limited in the literature. However, due to its high portability, UROMAN can help scale the process of transliteration massively and as such benefit low-resource scenarios and wider adaptation of mPLMs. The main idea, as hinted in §1, is to (learn to) map

any UTF-8 character to the Latin script, without the use of any external language-specific dictionaries (see Hermjakob et al. (2018) for technical details).

**Cross-Lingual Transfer to Low-Resource Languages.** Parameter-efficient and modular fine-tuning methods (Pfeiffer et al., 2023) such as adapters (Houlsby et al., 2019; Pfeiffer et al., 2020b) have been used for cross-lingual transfer, putting a particular focus on enabling transfer to low-resource languages and scenarios, including languages with scripts 'unseen' by the base mPLM (Pfeiffer et al., 2021). Adapters are small lightweight components stitched into the base mPLM, and then trained for particular languages and tasks while keeping the parameters of the original mPLM frozen. This circumvents the issues of catastrophic forgetting and interference (McCloskey and Cohen, 1989) within the mPLM, and allows for extending its reach also to unseen languages (Pfeiffer et al., 2021; Ansell et al., 2021).

For our main empirical analyses, we adopt a state-of-the-art modular method for cross-lingual transfer: MAD-X (Pfeiffer et al., 2020b). In short, MAD-X is based on language adapters (LA), task adapters (TA), and invertible adapters (INV). While LAs are trained for specific languages relying on masked language modeling, TAs are trained with high-resource languages relying on task-annotated data and task-specific objectives. At inference, the source LA is replaced with the target LA while the TA is kept. In order to do parameter-efficient learning for the token-level embeddings across different languages and to deal with the vocabulary mismatch between source and target languages, Pfeiffer et al. (2020b) also propose INV adapters: they are placed on top of the embedding layer and their inverses precede the output embedding layer.[3] We adopt the better-performing MAD-X 2.0 setup (Pfeiffer et al., 2021) where the adapters in the last Transformer layer are dropped at inference.[4]

## 3 Experiments and Results

As the main means of analyzing the impact of transliteration in general and UROMAN-based romanization in particular, we train different variants of language adapters within the MAD-X framework, based on transliterated and non-transliterated

---

[2]Code and data available at https://github.com/UKPLab/emnlp23_romanization_based_adaptation

[3]They are trained together with the LAs while the rest of the mPLM is kept frozen.

[4]We refer the reader to the original papers for further technical details regarding the MAD-X framework.

| Language | Family | Script | # Sentences |
|---|---|---|---|
| Bhojpuri (bh) | Indo-European | Devanagari | 35,983 |
| Buryat (bxr) | Mongolic | Cyrillic | 41,692 |
| Erzya (myv) | Uralic | Cyrillic | 42,575 |
| Meadow Mari (mhr) | Uralic | Cyrillic | 144,529 |
| Min Dong (cdo) | Sino-Tibetan | Chinese | 33,978 |
| Mingrelian (xmf) | Kartvelian | Georgian | 63,032 |
| Sindhi (sd) | Indo-European | Arabic | 86,176 |
| Sorani Kurdish (ckb) | Indo-European | Arabic | 459,475 |
| Uyghur (ug) | Turkic | Arabic | 149,813 |
| Amharic (am) | Afro-Asiatic | Geʾez | 88,320 |
| Divehi (dv) | Indo-European | Thaana | 34,779 |
| Khmer (km) | Austroasiatic | Khmer | 139,704 |
| Sinhala (si) | Indo-European | Sinhala | 219,866 |
| Tibetan (bo) | Sino-Tibetan | Tibetan | 131,362 |

Table 1: Languages with their ISO 639-3 codes used in our evaluation, along with their script, language family, and number of sentences available for pretraining. The dashed line separates languages with unseen scripts, placed in the bottom part of the table.

versions of target language data, outlined here.

**Variants with Non-Transliterated Data.** For the **Non-Trans$_{\text{LA+INV}}$** variant, we train LAs and INV adapters together. This variant serves to examine the extent to which mPLMs can adapt to unseen languages without any vocabulary extension.[5] We compare this to **Non-Trans$_{\text{LA+Emb}_{\text{Lex}}}$**, which trains a new tokenizer for the target language (Pfeiffer et al., 2021): the so-called 'lexically overlapping' tokens are initialized with mPLM's trained embeddings, while the remaining embeddings are initialized randomly. All these embeddings (Emb$_{\text{Lex}}$) are fine-tuned along with LAs.

**Variants with Transliterated Data.** We evaluate a **Trans$_{\text{LA+INV}}$** variant, which uses the same setup as Non-Trans$_{\text{LA+INV}}$ but now with transliterated data. We again note that in this efficient setup, we do not extend the vocabulary size, and use the fewest trainable parameters. In the **Trans$_{\text{LA+mPLM}_{\text{ft}}}$** variant, we train LAs along with fine-tuning the pretrained embeddings of mPLM (mPLM$_{\text{ft}}$). This further enhances the model capacity by fine-tuning the embedding layer instead of using invertible adapters.[6] For both variants, transliterated data can be produced via different transliterators: (i) language-specific ones; (ii) the ones from 'language siblings' (e.g., using a Georgian transliterator for Mingrelian), or (iii) UROMAN.

---

[5]Since LAs without INV typically perform worse than with INV (Pfeiffer et al., 2020b), also confirmed in our preliminary experiments, we do not ablate to the setup without INV.

[6]We do not have this setup for non-transliterated data since, for languages with unseen scripts, most of the tokens are replaced by the generic 'UNK' token, and fine-tuning embeddings hardly benefit downstream performance.

## 3.1 Experimental Setup

**Data, Languages and Tasks.** Following Pfeiffer et al. (2021), we select mBERT as our base mPLM. We experiment with $14$ typologically diverse low-resource languages that are not part of mBERT's pretraining corpora, with $5/14$ languages written in distinct scripts (see Table 1 for details). For LA training, we use Wikipedia dumps for the target languages, which we also transliterate (using different transliterators). Evaluation is conducted on two standard cross-lingual transfer tasks in zero-shot setups: **1)** the WikiAnn NER dataset (Pan et al., 2017) with the train, dev, and test splits from (Rahimi et al., 2019); **2)** for dependency parsing, we rely on the UD Dataset v2.7 (Nivre et al., 2020).

**LAs and TAs.** English is the source language in all experiments, and is used for training TAs. The English LA is obtained directly from Adapterhub.ml (Pfeiffer et al., 2020a), LAs and embeddings (when needed) are only trained for target languages.

Finally, for the Non-Trans$_{\text{LA+Emb}_{\text{Lex}}}$ variant, we train a WordPiece tokenizer on the target language data with a vocabulary size of $10$K.

**Training of Language and Task Adapters.** We train all the language adapters for $50$ epochs or $\sim 50K$ update steps based on the corpus size. The batch size is set to $64$ and the learning rate is $1e-4$.

We train English task adapters following the setup from (Pfeiffer et al., 2020b). For NER, we directly obtain the task adapter from Adapterhub.ml which is trained with a learning rate of $1e-4$ for $10$ epochs. For DP, we train a Transformer-based (Glavaš and Vulić, 2021) biaffine attention dependency parser (Dozat and Manning, 2017). We use a learning rate of $5e-4$ and train for $10$ epochs as in (Pfeiffer et al., 2021).

All the reported results in both tasks (NER and DP) are reported as averages over $6$ random seeds. All the models have been trained on A100 or V100 GPUs. None of the training methods consumed more than 36 hours. As the main means of analyzing the impact of transliteration in general and UROMAN-based romanization in particular, we train different variants of language adapters within the MAD-X framework, based on transliterated and non-transliterated versions of target language data, outlined here.

## 3.2 Results and Discussion

**UROMAN versus Other Transliterators and Transliteration Strategies.** In order to estab-

| Task | Transliterator | am | ar | ka | ru | hi | sd | **avg** |
|---|---|---|---|---|---|---|---|---|
| NER (Macro F1) | UROMAN | 25.6 | 24.8 | 61.4 | 66.5 | 48.6 | 35.3 | **43.7** |
| | Other | 25.5 | 23.7 | 57.3 | 63.9 | 56.7 | 35.9 | **43.8** |
| UD (UAS / LAS) | UROMAN | 36.1 / 6.6 | 33.0 / 19.8 | - | 47.3 / 32.4 | 33.8 / 17.8 | - | **37.5 / 19.1** |
| | Other | 29.9 / 5.4 | 32.6 / 19.9 | - | 45.0 / 19.9 | 33.2 / 17.9 | - | 35.2 / 15.8 |

Table 2: Comparison of UROMAN with language-specific transliterators.

| Method | Seen Script | | | | | | | Unseen Script | | | | | **avg** |
|---|---|---|---|---|---|---|---|---|---|---|---|---|---|
| | bh | cdo | ckb | mhr | sd | ug | xmf | am | bo | dv | km | si | |
| UROMAN | 32.59 | 27.34 | **67.73** | 64.68 | **35.33** | 28.10 | 52.58 | 25.69 | 35.95 | 29.99 | 41.76 | 31.83 | 26.89 |
| BORROW | **53.42 (hi)** | - | 12.46 (ar) | 45.86 (ru) | 16.79 (ar) | 12.85 (ar) | 24.77 (ru) | - | - | - | - | - | - |
| RAND | 25.42 | 19.51 | 53.55 | 42.02 | 27.20 | 25.18 | 35.82 | 18.00 | 18.95 | 21.19 | 32.75 | 20.01 | 21.59 |

Table 3: Comparison of various transliteration strategies on the NER task (Macro-F1).

| Variant | Seen Script | | | | | | | Unseen Script | | | | | **avg** |
|---|---|---|---|---|---|---|---|---|---|---|---|---|---|
| | bh | cdo | sd | xmf | mhr | ckb | ug | am | bo | dv | km | si | |
| Non-Trans$_{LA+INV}$ | **55.14** | 24.19 | 31.31 | 49.74 | **70.31** | 45.54 | **33.53** | 3.26 | 19.86 | 18.72 | 13.81 | 23.14 | 18.39 |
| Trans$_{LA + INV}$ | 32.59 | 27.34 | 35.33 | 52.58 | 64.68 | 67.73 | 28.10 | 25.69 | 35.95 | 29.99 | 41.76 | 31.83 | 26.89 |
| Non-Trans$_{LA+Emb_{Lex}}$ | **60.00** | 28.91 | **42.47** | 51.99 | 61.05 | **79.12** | **50.42** | **47.60** | **40.96** | 31.21 | **53.94** | **45.89** | **49.01** |
| Trans$_{LA + mPLM_{ft}}$ | 49.05 | **36.92** | 39.16 | **57.99** | 69.85 | 73.92 | 33.43 | 37.09 | 33.82 | **40.40** | 52.39 | 45.24 | 47.44 |

Table 4: Results (Macro-F1 scores) on WikiAnn NER averaged over 6 random seeds.

| Variant | Seen Script | | | | Unseen Script | |
|---|---|---|---|---|---|---|
| | bh | myv | ug | bxr | am | **avg** |
| Non-Trans$_{LA+INV}$ | **28.46 / 11.53** | 45.28 / 26.27 | **33.44 / 15.28** | **39.75 / 19.77** | 19.08 / 1.85 | 33.20 / 10.81 |
| Trans$_{LA + INV}$ | 25.12 / 10.17 | **45.74 / 26.64** | 32.30 / 15.10 | 37.92 / 17.23 | **36.07 / 7.58** | **35.43 / 12.41** |
| Non-Trans$_{LA+Emb_{Lex}}$ | 26.68 / 10.10 | **48.34 / 25.34** | 41.20 / **22.81** | **39.51** / 16.02 | 36.47 / 8.39 | **38.44** / 12.20 |
| Trans$_{LA + mPLM_{ft}}$ | **28.04** / 11.13 | 41.97 / 20.29 | **50.89** / 16.56 | 35.03 / **20.29** | **39.10 / 9.00** | **39.01 / 14.65** |

Table 5: Results (UAS / LAS scores) in the DP task with UD, averaged over 6 random seeds.

lish the utility of UROMAN as a viable transliterator, especially for low-resource languages, we compare its performance with transliteration options using the Trans$_{LA + INV}$ setup as the most efficient scenario. First, we compare UROMAN with language-specific transliterators available for selected languages: *amseg* (Yimam et al., 2021) for Amharic, *ai4bharat-transliteration* (Madhani et al., 2022) for Hindi and Sindhi, *lang-trans* for Arabic, and *transliterate* for Russian and Georgian[7]. The transliterators used in this work are outlined in Table 6. The results are provided in Table 2. On average, UROMAN performs better or comparable to the language-specific transliterators. This provides justification to use UROMAN for massive transliteration at scale.

Second, we compare UROMAN to two other transliteration strategies. (i) *BORROW* refers to borrowing transliterators from languages within the same language family and written in the same script.[8] Since building transliterators are costly, this gives us an estimate of whether it is possible to rely on the related transliterators when we do not have a language-specific one at hand. (ii) *RAND*

refers to a random setting where we associate any non-ASCII character with any ASCII character, giving us an estimate of whether we actually need knowledge of the language to build transliterators. The results are provided in Table 3: UROMAN is largely and consistently outperforming both BORROW and RAND, where the single exception is BORROW (from Hindi to Bhojpuri). Surprisingly, RAND also yields reasonable performance and on average even outperforms the Non-Trans$_{LA+INV}$ variant with non-transliterated data (21.59 vs 18.39 in Table 4 later). This provides further evidence towards the utility of transliteration in general and UROMAN-based romanization in particular to assist and improve language adaptation.

**Performance on Low-Resource Languages** is summarized in Table 4 and Table 5.[9] We note that Trans$_{LA+INV}$ outperforms Non-Trans$_{LA+INV}$ for all the languages with unseen scripts, and achieves that with huge margins ($\sim$ 8-22 points for NER and $\sim$ 17 points in UAS scores). We observe similar trends for some of the languages with seen scripts such as Min Dong (cdo), Sindhi (sd), Mingrelian

---
[7]For reproducibility, the links to the language-specific transliterators are available in Appendix A

[8]E.g., a Hindi transliterator can be borrowed for Bhojpuri since the two are related and written in Devanagari.

[9]We also compare the performance of these methods with the standard cross-lingual transfer setup for finetuning an out-of-the-box mBERT for languages with unseen scripts in Appendix B. All the adapter-based methods massively outperform an out-of-the-box mBERT in this scenario.

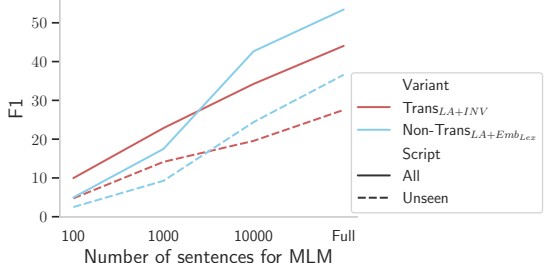

Figure 2: Sample efficiency in the NER task.

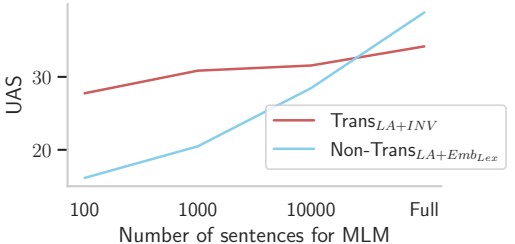

Figure 3: Sample efficiency in the DP task.

(xmf) on NER tasks and Erzya (myv) on DP. The less efficient $\text{Trans}_{LA+mPLM_{ft}}$, as expected, further improves the performance for all the languages except for Tibetan (bo).[10] Non-$\text{Trans}_{LA+Emb_{Lex}}$, however, now outperforms UROMAN-based methods for a majority of the languages. This observation can be attributed to various factors related to mBERT's tokenizer, and we provide an in-depth analysis later in Appendix C. Nonetheless, we observe strong and competitive performance of $\text{Trans}_{LA+mPLM_{ft}}$ in both tasks, again indicating that more attention should be put on transliteration-based language adaptation in future work.

**Sample Efficiency.** Finally, we simulate a few-shot setup to study the effectiveness of using transliterated versus non-transliterated data in data-scarce scenarios. For NER, we evaluate performance on all the languages and on languages with unseen scripts; for DP, we evaluate on all the languages. Figure 2 indicates that $\text{Trans}_{LA+INV}$ on average performs better than all the other methods at sample sizes 100 (i.e., 100 sentences in the target language) and $1,000$. However, from $10,000$ sentences onward, Non-$\text{Trans}_{LA+Emb_{Lex}}$ takes the lead. We observe similar trends in the DP task (see Fig 3). This establishes the utility of transliteration for (extremely) low-resource scenarios.

---

[10]For Tibetan, longer words are composed using shorter words separated by *tsek* (".") which is not a valid space delimiter for the mBERT tokenizer; the number of produced subwords is thus much higher than for the other languages.

# 4 Conclusion

In this work, we have systematically analyzed and confirmed the potential of romanization, implemented via the UROMAN tool, to help with adaptation of multilingual pretrained language models. Given (i) its broad applicability and (ii) strong performance overall and for languages with unseen scripts, we hope our study will inspire more work on transliteration-based adaptation.

## Limitations

In this paper, we work with UROMAN (Hermjakob et al., 2018) which is an unsupervised romanization tool. While it is an effective tool for romanization at scale, it still has potential drawbacks. Since it is only based on lexical substitution, its transliterations may not semantically or phonetically align with the source content and may differ from transliterations preferred by native speakers. Moreover, UROMAN is not invertible—as we have highlighted—and may thus be less appealing when text in the original script needs to be exactly reproduced. Our proposed method, while it is parameter-efficient and effective—particularly for low-resource languages—still underperforms language-specific tokenizer-based non-transliteration methods. Future work may focus on developing an improved and more efficient tokenizer for transliteration-based methods as we highlight in the Appendix.

While there is now a growing body of available evaluation resources for low-resource languages (Ebrahimi et al., 2022; Mhaske et al., 2023; Winata et al., 2023, *among others*), our final selection of tasks, resources and languages has been driven and constrained by the specific concrete goal of our short paper: studying and evaluating if and how transliteration/romanization can help with adaptation of languages with scripts unseen by the pretrained multilingual language model. We thus closely follow the experimental setup of Pfeiffer et al. (2021) which used the same set of tasks and languages with unseen scripts.

Finally, romanization can be seen as a step towards providing more universal, or rather language-agnostic, input text representation. Full-fledged comparisons against other approaches that aim to strike language independence at the input or feature level, such as byte-level models (e.g., ByT5 (Xue et al., 2022)) and pixel-based models (e.g., PIXEL (Rust et al., 2023)) go beyond

the scope of this particular work, but we point out to this as a very interesting future research avenue. Moreover, the integration of these language-agnostic representations with 'romanization'-based approaches might yield additional benefits, and should also be attested in future research.

## Acknowledgements

This work has been funded by the German Research Foundation (DFG) as part of the Research Training Group KRITIS No. GRK 2222. The work of Ivan Vulić has been supported by a personal Royal Society University Research Fellowship (no 221137; 2022-).

We thank Indraneil Paul, Yongxin Huang, Ivan Habernal, and Massimo Nicosia for their valuable feedback and suggestions on a draft of this paper.

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

Judit Ács. Exploring bert's vocabulary. *Blog Post*.

## A  Transliterators in Evaluation

Besides UROMAN, we also employ various language-specific transliterators which are publicly available. We list them in Table 6.

## B  Performance comparison of mBERT

We adapt the standard cross-lingual transfer setup for mBERT. The model is finetuned on the task data for a source language (high-resource) and is used to perform inference on the target language (low-resource). We report the performance comparison of the standard cross-lingual transfer setup for mBERT on the NER task for languages with unseen scripts with the adapter-based methods in Table 7. We observe that the adapter-based methods outperform mBERT by huge margins.

## C  Further Analyses

Following previous work (Ács; Rust et al., 2021; Moosa et al., 2023), we further analyze tokenization quality of the mBERT tokenizer using the following established metrics: **1) % of "UNK"s** measures the % of "UNK" tokens produced by the tokenizer, and our aim is to compare their rate before and after transliteration; **2) fertility** measures the number of subwords that are produced per tokenized word; **3) proportion of continued subwords** measures the proportion of words for which the tokenized word is split across at least two subwords (denoted by the symbol ##).

From the results summarized in Figure 4, it is apparent that transliteration drastically reduces % of UNKs. However, mBERT's tokenizer underperforms as compared to monolingual tokenizers based on fertility and the proportion of continued subwords (Rust et al., 2021). Transliteration performs better for some languages where the quality of the mBERT tokenizer is similar to the monolingual tokenizer such as for dv, km, and cdo. On the other hand, transliteration methods perform worse on languages where the quality of the underlying mBERT tokenizer is relatively poor.

In order to test the hypothesis that the tokenizer quality might be the principal reason for the performance gap for the transliteration-based methods in comparison to the non-transliteration based methods, we carried out an additional experiment. For the experiment, we adapt the Non-Trans$_{LA+Emb_{Lex}}$ to operate on transliterated data, and call this variant Trans$_{LA+Emb_{Lex}}$. Here, we train a new tokenizer on the transliterated data and initialize lexically overlapping embeddings with mBERT's pretrained embeddings.

We plot the performance in Figure 5. The new method, Trans$_{LA+Emb_{Lex}}$ now outperforms the non-transliteration-based variant on 8/12 languages and also on average. Consequently, this validates our hypothesis and is in line with the previous work (Moosa et al., 2023). However, we found a drop in performance in the case of mhr (-10.71) and cdo (-10.14) when compared to Trans$_{LA + mPLM_{ft}}$. These drops may be attributed to the lower degree of lexical overlap with mBERT's vocabulary, and consequently a higher number of randomly initialized embeddings for those target languages.

| Transliterator | Used for languages | Available at |
|---|---|---|
| UROMAN | **All** | github.com/isi-nlp/uroman |
| *amseg* | am | pypi.org/project/amseg/ |
| *transliterate* | ru, ka | pypi.org/project/transliterate/ |
| *ai4bharat-transliteration* | hi, sd | pypi.org/project/ai4bharat-transliteration/ |
| *lang-trans* | ar | pypi.org/project/lang-trans/ |

Table 6: Transliterators used in this work.

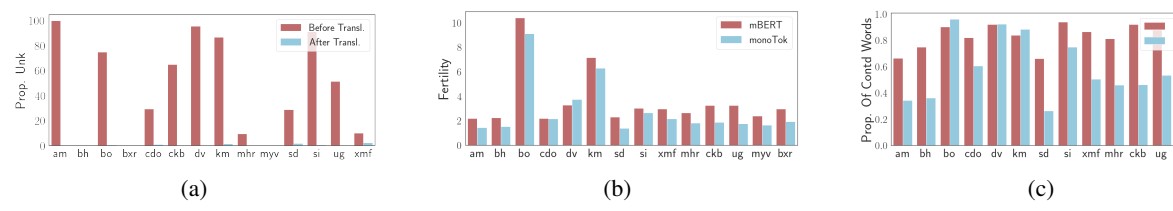

(a)  (b)  (c)

Figure 4: Tokenizer quality analysis. a) % of UNKs before and after transliteration, b) Fertility, and c) Proportion of continued subwords for mBERT vs monolingual tokenizer.

|  | **Unseen Scripts** | | | | |
|---|---|---|---|---|---|
| **Variant** | am | bo | dv | km | si |
| mBERT | 0.91 | 17.40 | 1.30 | 10.71 | 2.50 |
| Non-Trans$_{LA+INV}$ | 3.26 | 19.86 | 18.72 | 13.81 | 23.14 |
| Trans$_{LA + INV}$ | 25.69 | 35.95 | 29.99 | 41.76 | 31.83 |
| Non-Trans$_{LA+Emb_{Lex}}$ | **47.60** | **40.96** | 31.21 | **53.94** | **45.89** |
| Trans$_{LA + mPLM_{ft}}$ | 37.09 | 33.82 | **40.40** | 52.39 | 45.24 |

Table 7: Performance Comparison of mBERT with adapter-based finetuning methods for unseen scripts on the NER task.

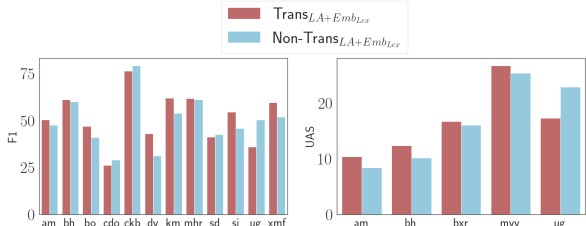

Figure 5: Comparison of Non-Trans$_{LA+Emb_{Lex}}$ with Trans$_{LA+Emb_{Lex}}$ on NER (left) and DP (right).