# OpenReview forum: "Romanization-based Large-scale Adaptation of Multilingual Language Models"
_EMNLP/2023/Conference — EMNLP 2023 Findings_

### Official Review · Reviewer_PTYq · 2023-07-26

**Soundness:** 3

**Excitement:**

3: Ambivalent: It has merits (e.g., it reports state-of-the-art results, the idea is nice), but there are key weaknesses (e.g., it describes incremental work), and it can significantly benefit from another round of revision. However, I won't object to accepting it if my co-reviewers champion it.

**Paper Topic And Main Contributions:**

This paper compares the zero-shot performance of MAD-X with mBERT on NER and UD parsing with and without transliterating non-ASCII characters to ASCII characters. MAD-X uses language and task adapter layers to adapt mBERT to different languages and tasks.

The experimental results indicate that

- the simple UROMAN transliteration method performs as good as language-specific transliterators on NER and better on UD parsing

- the UROMAN-based system outperforms regular systems whose vocabulary is not extended.

- the UROMAN-based system performs similar to regular systems whose vocabulary _is_ extended.

- the UROMAN-based system with extended vocabulary outperforms regular systems with vocabulary extension.

-  the UROMAN-based system is particularly useful when low-resource training data is very scarce.

**Reasons To Accept:**

The paper shows that UROMAN is a good alternative to language-specific transliteration systems for NLP purposes.
It also shows that transliteration-based handling of foreign scripts gives good results.

**Reasons To Reject:**

All transliteration-based systems which perform well modify the embedding table. Despite using adapters, the original mBERT system is therefore not preserved. Hence there is no reason not to finetune _all_ mBERT parameters and results for such a finetuning baseline should be presented as well.

The performance difference between Non-Trans LA+Emb_Lex and Trans LA + mPLM_ft is small on average. At the same time, the results for individual languages vary greatly. This suggests that the differences might be insignificant. The authors should therefore do a significance test.
It was not clear to me whether the results have been obtained from single runs or whether they are averages over several runs. The latter could reduce the above-mentioned variance in the results.

**Reproducibility:**

3: Could reproduce the results with some difficulty. The settings of parameters are underspecified or subjectively determined; the training/evaluation data are not widely available.

**Reviewer Confidence:**

4: Quite sure. I tried to check the important points carefully. It's unlikely, though conceivable, that I missed something that should affect my ratings.

**Typos Grammar Style And Presentation Improvements:**

Figure 2: The image legend hides a part of the graph.

unclear: "RAND refers to a random setting where we associate any non-ASCII character with any ASCII character"
Did you randomly select a single ASCII character for each non-ASCII character?

unclear definition: "proportion of continued subwords measures the proportion of words for which the tokenized word is split across at least two subwords"

---

> ### Author Rebuttal · Authors · 2023-08-28
>
> We would like to thank the reviewer for their useful feedback and suggestions.
>
> # **Reasons**
> #### $R1$. **“All transliteration-based systems which perform well modify the embedding table. Despite using adapters, the original mBERT system is therefore not preserved. Hence there is no reason not to finetune all mBERT parameters and results for such a finetuning baseline should be presented as well."**
>
> $Ans$. Our decision to finetune the embedding layer while keeping the transformer weights frozen is motivated by the goal of preserving the transferability of transformer blocks across languages. Full finetuning of the transformer blocks on source or target tasks can lead to a drop in transfer performance, as indicated in previous research, such as the "MAD-X" study [1]. This drop in transferability occurs because the specificity introduced through full finetuning hampers the model's ability to generalize effectively across diverse languages and linguistic characteristics.
>
> By focusing on finetuning the embedding layer, we aim to achieve a balance between language specificity and cross-lingual transfer. This approach aligns with the principles demonstrated in the "XQuAD" paper [2], where finetuning the embedding layer allowed monolingual models to become multilingual, providing enhanced language coverage with manageable adjustments.
>
> From a parameter-efficiency perspective, finetuning only the embedding layer offers significant advantages. Making the embedding layer language-specific maintains the benefits of a shared transformer architecture while adapting the model to language-specific linguistic features.
>
> The concept of posthoc analysis to identify changed embeddings and then map them to respective target languages further enhances the efficiency of the proposed setup. This approach minimizes the computational cost associated with updating embeddings, making it practical for production-level deployment.
>
> #### $R2$. **“The performance difference between Non-Trans LA+Emb_Lex and Trans LA + mPLM_ft is small on average. At the same time, the results for individual languages vary greatly. This suggests that the differences might be insignificant. The authors should therefore do a significance test. It was not clear to me whether the results have been obtained from single runs or whether they are averages over several runs. The latter could reduce the above-mentioned variance in the results.”**
>
> $Ans$. All the reported results in our paper have been obtained by averaging over six random seeds, as indicated in the captions of Table 3 and Table 4. We will highlight this information more prominently in the paper following your suggestion.
>
> # **Typos**
> #### $T1$. **“Figure 2: The image legend hides a part of the graph.”**
>
> $Ans$. We will improve the figure, apologies for the oversight.
>
> #### $T2$. **"unclear: "RAND refers to a random setting where we associate any non-ASCII character with any ASCII character" Did you randomly select a single ASCII character for each non-ASCII character?"**
>
> $Ans$. We considered all the ASCII characters and randomly assigned any ASCII character to any Non-ASCII character. It is a many-to-many mapping.
>
> #### $T3$. **unclear definition: "proportion of continued subwords measures the proportion of words for which the tokenized word is split across at least two subwords"**
>
> $Ans$. The metric termed “Proportion of continued words” measures proportion of words in a particular language which gets split into more than two subwords. In case of mBERT, these subwords are preceded by ## symbol. For more information we refer the reader to [3]. We will clarify this in the paper.
>
> ### **References**
>
> [1] “MAD-X: An Adapter-Based Framework for Multi-Task Cross-Lingual Transfer” (Pfeiffer et. al, EMNLP 2020)
> [2] “On the Cross-lingual Transferability of Monolingual Representations” (Artetxe et. al, ACL 2020)
> [3] “How Good is Your Tokenizer? On the Monolingual Performance of Multilingual Language Models” (Rust et al., ACL 2021)

---

### Official Review · Reviewer_oGZa · 2023-08-04

**Soundness:** 3

**Excitement:**

3: Ambivalent: It has merits (e.g., it reports state-of-the-art results, the idea is nice), but there are key weaknesses (e.g., it describes incremental work), and it can significantly benefit from another round of revision. However, I won't object to accepting it if my co-reviewers champion it.

**Paper Topic And Main Contributions:**

In this paper, the authors aim to utilise transliteration to improve cross-lingual transfer of languages not written with the Latin script, especially targetting low-resource and unseen languages. This romanisation avoids the extension of the original model vocabulary and can help to better leverage existing model information thanks to overlapping vocabulary items. The authors employ **UROMAN**, a universal romanisation tool and demonstrate in many cases marked performance benefits on a diverse set of 14 low-resource languages on NER and dependency parsing. Additionally, they demonstrate the sample efficiency of their methodology, which is promising for achieving performance gains over low-resource languages.

Despite gaps in the literature review and the lack of comparison with other methods, such as other universal romanisation frameworks or byte-based tokenization, the paper is informative all-in-all.

Some minor notes:
* The only BORROW setup that outperforms both UROMAN and RAND when it comes to transliteration is when transliterating Bhojpuri using a Hindi tool. It might be nice to mention that this is probably because the two languages are closely related unlike for instance Meadow Mari and Russian.
* When describing why performance on Tibetan drops when using their Trans_LA+INV setup as compared to Non-Trans_LA+INV, the authors give a nice explanation related to the word separator character in Tibetan. They also make the claim that the number of produced subwords becomes higher for Tibetan than for other languages because of this. It might be nice to show some graph that demonstrates this.
* When mentioning the correlation in Appendix E between the performance reached via transliteration and the quality of the mBERT tokenizer, it would be nice if a figure could demonstrate this for the reader as well.

**Questions For The Authors:**

A: When comparing monolingual transliteration tools borrowed from languages written with the same script, UROMAN, and a random setup, the random setup achieves a surprisingly high performance. Do the authors have any idea why this might be the case?

B: Do the authors think that high fertility values or a high proportion of continued subwords necessarily indicate that the tokenizer performs worse for the target language? Would this be the case even for morphologically complex languages?

**Reasons To Accept:**

* The paper is addressing an important and pervasive issue in multilingual NLP.
* There is some robust analysis demonstrating the sample efficiency of the method as well as explaining the cases where it might underperform.
* The paper compares a large number of setups and the inclusion of these seems well-motivated.
* The authors achieve definite improvements over other methods. Their results are made more robust by the fact that they represent the average over 6 random seeds.

**Reasons To Reject:**

* The authors make no reference to byte-based (ByT5; Xue et al., 2021) and/or pixel-based (PIXEL; Rust et al., 2022) tokenization methods, or other universal transliteration tools. The paper would benefit from a reference to at least byte-based and pixel-based methods and some justification why romanisation might be more appropriate to use. This justification could also include some performance comparison.
* The authors do not compare their results with a baseline of using an out-of-the-box mBERT model without any language adapters. The inclusion of this (which would likely underperform all the other methods presented) would make their results that much more robust.
* Since the UROMAN transliteration is often irreversible, there is a problem of how useful this method might be for real-world use. However, the authors are aware of this and mention it in their limitations section.
* There are no details on the training data sizes the language adapters were trained on.

**Reproducibility:**

4: Could mostly reproduce the results, but there may be some variation because of sample variance or minor variations in their interpretation of the protocol or method.

**Reviewer Confidence:**

3: Pretty sure, but there's a chance I missed something. Although I have a good feel for this area in general, I did not carefully check the paper's details, e.g., the math, experimental design, or novelty.

**Typos Grammar Style And Presentation Improvements:**

* It might be better to have the languages used listed in the main body of the paper and not just in the appendix.

---

> ### Author Rebuttal · Authors · 2023-08-28
>
> We would like to thank the reviewer for their useful and detailed feedback. Below are our justifications for their concerns.
>
> # **Reasons**
>
> #### $R1$. **"The authors make no reference to byte-based (ByT5; Xue et al., 2021) and/or pixel-based (PIXEL; Rust et al., 2022) tokenization methods, or other universal transliteration tools. The paper would benefit from a reference to at least byte-based and pixel-based methods and some justification why romanisation might be more appropriate to use. This justification could also include some performance comparison.”**
>
> $Ans$. Byte-based tokenization, such as ByT5 [1], and pixel-based tokenization, such as PIXEL [2], can mitigate the issue of unknown-token fallbacks for unseen characters. However, the representations of these unseen tokens remain uninformative, which may limit the model's ability to generalize effectively to new scripts and languages. In contrast, transliteration provides a means of converting unseen characters into familiar Latin characters and subword, enabling the model to leverage existing knowledge about Latin script characters and subwords. This inherently offers more informative zero-shot information, as it allows the model to draw on the relationships between scripts and languages while handling unseen tokens. We will make these connections and (dis)similarities clearer in the paper by adding a discussion on the advantages of romanization.
>
> $R2$. **“The authors do not compare their results with a baseline of using an out-of-the-box mBERT model without any language adapters. The inclusion of this (which would likely underperform all the other methods presented) would make their results that much more robust.”**
>
> $Ans$.  A significant portion of the languages we evaluated feature scripts that are entirely unseen during mBERT's pretraining. Consequently, employing an out-of-the-box mBERT model without any language adapters or additional adaptation is not a viable option for accurately representing these scripts. The inherent inability of mBERT to handle these scripts has been demonstrated in prior research, as highlighted in [3] and [4]. Our preliminary experiments align with the aforementioned research findings, showing that full finetuning of mBERT without specialized adaptation performs poorly on languages with unseen scripts. For instance, our initial experiments validated this trend, with Amharic achieving $0.9$ F1 on WikiAnn, Tibetan achieving $17.4$, and Khmer achieving $10.7$. We will incorporate the performance metrics achieved by an out-of-the-box mBERT model without any language adapters as part of our main results.
>
> $R3.$ **“There are no details on the training data sizes the language adapters were trained on.”**
>
> $Ans$. We apologize for the oversight in not providing the training data sizes for the language adapters in the initial version of our paper. The number of sentences used for each language adapter is as follows:
>
> * Amharic ($88320$)
> * Bhojpuri ($35983$)
> * Tibetan ($131362$)
> * MinDong ($33978$)
> * Divehi ($34779$)
> * Khmer ($139704$)
> * Sindhi ($86176$)
> * Sinhala ($219866$)
> * Mingrelian ($63032$)
> * Meadow Mari ($144529$)
> * Sorani Kurdish ($459475$)
> * Buryat ($41692$)
> * Uyghur ($149813$)
>
> We will include a section that explicitly presents the training data sizes for each language adapter, ensuring that this information is readily available to readers.
>
> # **Questions**
> #### $Q1$. **“When comparing monolingual transliteration tools borrowed from languages written with the same script, UROMAN, and a random setup, the random setup achieves a surprisingly high performance. Do the authors have any idea why this might be the case?”**
>
> $Ans$. One crucial factor contributing to the observed behavior is the composition of mBERT's vocabulary. A substantial portion (77%) of mBERT's vocabulary consists of Latin characters, which are common across various languages. Consequently, when characters in a non-Latin script are transformed to Latin characters in the random setup, the number of subwords generated tends to decrease. This reduction in subword count can lead to better performance due to a closer match between the tokenization of the training data and the pretrained vocabulary. However, the situation differs when dealing with non-Latin scripts that are entirely unseen during pretraining. In these cases, the transliteration process introduces a large number of subwords due to the unique characteristics of these scripts. The abundance of subwords generated in the process of transliteration can lead to suboptimal performance as the subword-to-vocabulary mapping becomes less informative.
>
> #### $Q2$. **"Do the authors think that high fertility values or a high proportion of continued subwords necessarily indicate that the tokenizer performs worse for the target language? Would this be the case even for morphologically complex languages?"**
>
> $Ans$. Subword fertility has been shown to have a strong correlation with the performance of multilingual models. This phenomenon has been explored and established in prior research, most notably in [5]. While subword fertility's impact is potentially more pronounced in languages with simpler morphologies, it is still a relevant consideration for morphologically complex languages. These languages can exhibit a wide range of word forms, affixes, and inflections, which may result in higher subword fertility. In such cases, the subword segmentation process becomes crucial for capturing the intricate morphological structure accurately.
>
> # **Typos**
> #### $T1$. **"It might be better to have the languages used listed in the main body of the paper and not just in the appendix."**
>
> $Ans$. Due to page limitations of a short paper, we were unable to include this information in the main body - plenty of prior work also listed the languages as part of the appendix. However, provided we are granted an additional page, we will add it to the main paper.
>
> ### **References**
> [1] “ByT5: Towards a Token-Free Future with Pre-trained Byte-to-Byte Models” (Xue et. al, TACL 2022)
> [2] “Language Modelling With Pixels” (Rust et. al, ICLR 2023)
> [3] “UNKs Everywhere: Adapting Multilingual Language Models to New Scripts" (Pfeiffer et al., EMNLP 2021)
> [4] "When Being Unseen from mBERT is just the Beginning: Handling New Languages With Multilingual Language Models" (Muller et al., NAACL 2021)
> [5] “How Good is Your Tokenizer? On the Monolingual Performance of Multilingual Language Models” (Rust et al., ACL 2021)

---

### Official Review · Reviewer_SUm1 · 2023-08-05

**Soundness:** 3

**Excitement:**

3: Ambivalent: It has merits (e.g., it reports state-of-the-art results, the idea is nice), but there are key weaknesses (e.g., it describes incremental work), and it can significantly benefit from another round of revision. However, I won't object to accepting it if my co-reviewers champion it.

**Paper Topic And Main Contributions:**

In this paper, the authors explore the use of universal romanization for extending pre-trained languages to unseen languages (for seen and unseen scripts). They use the UROMAN romanization scheme based on UNICODE standard documentation for the same. They evaluate 2 tasks (NER and DP) in a zero-shot cross-lingual setup. Through the experiments and studies, the paper concludes that UROMAN romanization is good or better than language-specific romanization schemes and other baselines.

**Questions For The Authors:**

- What are the similar languages used for each language for BORROW strategy?

**Reasons To Accept:**

- The paper shows that a universal romanization scheme is good for adapting PLMs to unseen languages, and is competitive with vocabulary adaptation and use of language specific romanization.
- Further, vocabulary adaptation using romanized data seems to further improve the model performance - which is an interesting result. This result is reported in the appendix, but it find it one of the interesting contributions of the paper and it is important enough to report in the main paper. Additionally, you could report results
- The paper is well-written, experiments are well-described and well-motivated.

**Reasons To Reject:**

- The evaluation is done on only two tasks. Both these are tasks that are to language-dependent to a fairly large extent. The good performance of NER due to romanization could possibly be also attributed to named entities in English having similar romanization. It would be intersting to understand the performance of universal romanization on semantic tasks like text classification, NLI or sentiment analysis.

- The novelty is the work is limited.  Some of findings confirm what is know in literature: the use of transliteration and vocabulary adaptation in low-resource scenarios is beneficial. The benefit of universal romanization with UROMAN for NLU is a cross-lingual setup is new, but UROMAN has been explored for multilingual NMT previously.

**Reproducibility:**

5: Could easily reproduce the results.

**Reviewer Confidence:**

5: Positive that my evaluation is correct. I read the paper very carefully and I am very familiar with related work.

---

> ### Author Rebuttal · Authors · 2023-08-28
>
> We would like to thank the reviewer for their useful feedback and suggestions. We address the concerns below:
> # Reasons
>    #### $R1$. **“The evaluation is done on only two tasks. Both these are tasks that are to language-dependent to a fairly large extent. The good performance of NER due to romanization could possibly be also attributed to named entities in English having similar romanization. It would be intersting to understand the performance of universal romanization on semantic tasks like text classification, NLI or sentiment analysis.”**
>
> **$Ans.$** We selected the WikiAnn and Dependency Parsing benchmarks due to their relevance in assessing model performance on languages and scripts that were not present in the pretraining data. These datasets have been used in prior research for similar reasons, as demonstrated by works such as [1] and [2].
> We acknowledge the importance of evaluating UROMAN's effectiveness on a variety of tasks, however, it's worth noting that identifying datasets that meet the criteria of featuring unseen languages or scripts while spanning multiple semantic tasks can be a challenging endeavor. To the best of our knowledge, there are no alternative datasets and additional tasks that align with our evaluation objectives.
>
> $R2$. **“The novelty is the work is limited. Some of findings confirm what is know in literature: the use of transliteration and vocabulary adaptation in low-resource scenarios is beneficial. The benefit of universal romanization with UROMAN for NLU is a cross-lingual setup is new, but UROMAN has been explored for multilingual NMT previously.”**
>
> **$Ans$.** While it's true that UROMAN has been explored previously in multilingual neural machine translation (NMT) tasks, our paper takes a fresh perspective on its application in the context of recent advancements in language adaptation methods. The field of NLP has seen significant advancements since the previous UROMAN explorations, including methods that involve extending the vocabulary or replacing the embedding layer. Our study investigates the compatibility and effectiveness of UROMAN within this modern landscape, demonstrating its relevance and applicability.
>
> Furthermore, one of our main contributions lies in comparing UROMAN, a universal transliterator that can be applied to any script, with labor-intensive and expensive human-curated transliteration tools. While previous research may have examined UROMAN in multilingual NMT, our focus is on the benefits and trade-offs of UROMAN in comparison to specialized transliterators in the context of large-scale language model adaptation. This comparison offers insights into the practicality and efficiency of UROMAN across diverse languages, including low-resource settings. Our paper's primary objective is to validate the performance of UROMAN within the framework of recent language adaptation methods, rather than merely replicating previous findings. Through rigorous experimentation, we establish that UROMAN-based transliteration can indeed achieve competitive performance, even when compared to carefully curated and expensive transliteration tools. These results emphasize the robustness and effectiveness of UROMAN in contemporary cross-lingual setups.
>
> # Questions
> #### $Q1$. **“What are the similar languages used for each language for BORROW strategy?”**
> **$Ans$.** Table 2 second row, outlines the languages used for the BORROW strategy in the parentheses. For Bhojpuri(bh), we used Hindi(hi) transliterator, Sorani Kurdish (ckb) - Arabic (ar), Meadow Mari (mhr) - Russian (ru), Sindhi (sd) - Arabic (ar), Uyghur (ug) - Arabic (ar) and Mingrelian (xmf) - Russian (ru). We will add a detailed table for these borrowed language mappings to the paper following your question.
>
> ### **References**
> [1] “UNKs Everywhere: Adapting Multilingual Language Models to New Scripts" (Pfeiffer et al., EMNLP 2021)
> [2] "When Being Unseen from mBERT is just the Beginning: Handling New Languages With Multilingual Language Models" (Muller et al., NAACL 2021)

---

### Meta-Review · Area_Chair_1quB · 2023-09-18

**Recommendation:** 3

**Metareview:**

This work proposes UROMAN to adapt multilingual pretrained language models (PLMs) through transliterating UTF-8 characters to Latin scripts. All authors agree that the task of adapting PLMs to unseen scripts is important and well-motivated. The strength of UROMAN is its generalizability to any new language with unseen scripts. The authors show that UROMAN adapts PLMs to new languages with unseen scripts as good as or even better than language-specific transliteration, and the choice of languages in the experimental setup is diverse enough (with runs over 6 different seeds).

The paper lacks some baseline results, such as the finetuning of the entire model for language adaptation. Note that in this current climate, finetuning the entire small models like mBERT is an extremely reasonable practice; adapters are particularly useful for models with billions of parameters, so the authors' argument against not finetuning the whole model as baseline comparison due to computational cost doesn't really hold in my opinion. There is also work that shows that finetuning the whole model is better than language adapters at mBERT model sizes [1, 2].  While it's mentioned in the rebuttal, the authors should also include the results for out-of-the-box mBERT results as well in the paper and the training details such as number of tokens for adaptation per language.

[1]: Ebrahimi, A., & Kann, K. (2021). How to Adapt Your Pretrained Multilingual Model to 1600 Languages. In ACL 2021.
[2]: Yong, Zheng-Xin, Hailey Schoelkopf, Niklas Muennighoff, Alham Fikri Aji, David Ifeoluwa Adelani, Khalid Almubarak, M. Saiful Bari et al. (2023) "Bloom+1: Adding language support to bloom for zero-shot prompting." In ACL 2023.

---

### Decision · Program_Chairs · 2023-10-07

**Decision:**

Accept-Findings

**Comment:**

This work proposes UROMAN to adapt multilingual pretrained language models (PLMs) through transliterating UTF-8 characters to Latin scripts. All authors agree that the task of adapting PLMs to unseen scripts is important and well-motivated. The strength of UROMAN is its generalizability to any new language with unseen scripts. The authors show that UROMAN adapts PLMs to new languages with unseen scripts as good as or even better than language-specific transliteration, and the choice of languages in the experimental setup is diverse enough (with runs over 6 different seeds).

The paper lacks some baseline results, such as the finetuning of the entire model for language adaptation. Note that in this current climate, finetuning the entire small models like mBERT is an extremely reasonable practice; adapters are particularly useful for models with billions of parameters, so the authors' argument against not finetuning the whole model as baseline comparison due to computational cost doesn't really hold in my opinion. There is also work that shows that finetuning the whole model is better than language adapters at mBERT model sizes [1, 2].  While it's mentioned in the rebuttal, the authors should also include the results for out-of-the-box mBERT results as well in the paper and the training details such as number of tokens for adaptation per language.

[1]: Ebrahimi, A., & Kann, K. (2021). How to Adapt Your Pretrained Multilingual Model to 1600 Languages. In ACL 2021.
[2]: Yong, Zheng-Xin, Hailey Schoelkopf, Niklas Muennighoff, Alham Fikri Aji, David Ifeoluwa Adelani, Khalid Almubarak, M. Saiful Bari et al. (2023) "Bloom+1: Adding language support to bloom for zero-shot prompting." In ACL 2023.